# Biodegradable Materials for Tissue Engineering: Development, Classification and Current Applications

**DOI:** 10.3390/jfb14030159

**Published:** 2023-03-16

**Authors:** Marcel Modrák, Marianna Trebuňová, Alena Findrik Balogová, Radovan Hudák, Jozef Živčák

**Affiliations:** Department of Biomedical Engineering and Measurement, Faculty of Mechanical Engineering, Technical University of Košice, 042 00 Košice, Slovakia; marianna.trebunova@tuke.sk (M.T.); alena.findrik.balogova@tuke.sk (A.F.B.); radovan.hudak@tuke.sk (R.H.); jozef.zivcak@tuke.sk (J.Ž.)

**Keywords:** biodegradability, scaffold, implant, bibliometrics, classification, biocompatibility

## Abstract

The goal of this review is to map the current state of biodegradable materials that are used in tissue engineering for a variety of applications. At the beginning, the paper briefly identifies typical clinical indications in orthopedics for the use of biodegradable implants. Subsequently, the most frequent groups of biodegradable materials are identified, classified, and analyzed. To this end, a bibliometric analysis was applied to evaluate the evolution of the scientific literature in selected topics of the subject. The special focus of this study is on polymeric biodegradable materials that have been widely used for tissue engineering and regenerative medicine. Moreover, to outline current research trends and future research directions in this area, selected smart biodegradable materials are characterized, categorized, and discussed. Finally, pertinent conclusions regarding the applicability of biodegradable materials are drawn and recommendations for future research are suggested to drive this line of research forward.

## 1. Introduction

The current growing importance of regenerative medicine and tissue engineering (TE) reflects the fact that bone metabolic and related diseases represent approximately 50% of all chronic diseases for people above the age of fifty. In addition, mechanical damage of bone often occurs because of an accident, required surgery and so forth. Bone defects or bone injuries caused by aging, traffic accidents, fractures, or bone tumor resection are among the serious problems in orthopedics because they cause major damage to health and lower the quality of life. Internal fixation is required for reconstructive surgery on fractured bone to maintain the anatomic reduction in the fragments and provide stability during the healing process. In the past, bone fractures were fixed by the methods of applying metal implants. To substitute the metal implants for internal fracture fixation, numerous biodegradable materials (BMs) were developed. Biodegradable implants are increasingly used in regenerative medicine and sports medicine [1]. To be used successfully for fracture fixation, BMs must have sufficient strength and not degrade too rapidly. In an ideal scenario, these implants would break down as the wound healed, transferring load gradually to the healing tissue.

Today’s regenerative medicine and tissue engineering are using a large portfolio of BMs, which are used largely as substitutes for damaged or missing hard tissue. Natural and synthetic biodegradable polymers and hydrolysable metals make the main components for the creation of temporary medical implants [2]. Recently, much attention has been paid to materials based on extracellular matrix (ECM) [3,4], which consist of proteins, glycosaminoglycans and glycoproteins [5]. There is no doubt that the development and application research of BMs has significantly intensified in the last decade, as evidenced by the growing number of publications in this area. The aim of this article is to characterize the most important groups of BMs with a focus on polymers as a dominant material used in TE. To provide a systematic overview of BMs, they are first categorized according to their origin and method of production. Subsequently, future research prospects of polymeric BMs are explored based on bibliometric analysis. Thereafter, current development of so-called smart BMs is analyzed and discussed. Finally, relevant conclusions and development trends in the field are outlined. 

## 2. Indications and Materials for Biodegradable Implants in Orthopedics

There are several typical clinical indications for the use of biodegradable implants in orthopedics, which are mostly used for fractures stabilization, osteotomy procedures, bone grafts and fusions [6,7]. Furthermore, they can be used in re-attachment of tendons, ligaments, meniscal cracks, and other tissue structures [8]. The most common indications for biodegradable implants in orthopedics include anterior cruciate ligament reconstruction, meniscus repair, and ankle fracture treatment [9]. The occurrence of clinical indications for biodegradable implants is comprehensively demonstrated in Figure 1.

As demonstrated in Figure 1, there are several clinical indications on the upper limb. In the shoulder area, biodegradable implants are applied in fracture fixation of the glenoid fossa and in shoulder lesions repair [10]. Other shoulder indications include reconstruction of various intra-articular and extra-articular abnormalities. Clinical indications for the arm include osteochondral fractures of head and epicondyles of the humerus [11]. Biodegradable implants are also used for fracture fixation of the radial head and radial neck [12,13,14]. Furthermore, these implants are used to treat fractures of metacarpals and phalanges, fixation of tendons and collateral ligaments, lunate and scaphoid fractures [15,16,17].

Further clinical indications include those which are related to the lower limb. In the knee region, anterior cruciate ligament reconstructions are treated with the use of biodegradable implants [10]. Biodegradable pins are appropriate for osteochondral fractures. In addition, meniscal tacks and biodegradable suture anchors allow for new ways to perform reconstruction after complicated knee injuries. Patella fractures can also be treated with these implants. The foot and ankle region also benefits from these innovative implants. Here implants made from BM are used for treatment of isolated fractures of the internal malleolus [18,19,20]. Further indications include fractures of metatarsals and phalanges, flake fracture of the talus and calcaneus [21,22,23]. In addition, Lisfranc’s dislocation, syndesmotic disruptions and osteotomies for hallux valgus are among the health conditions that can be treated with biodegradable implants.

Traditionally, non-degradable metals such as inert stainless steel, titanium and its alloys, and cobalt-chromium alloys were used for internal fixation of fractured bones and joints [24]. These materials used to lack bone ingrowth in the scaffold and cause that the scaffold did not respond to changes in bone topology [25]. However, this issue is solvable when scaffold surface has porous structure or coatings that promote bone cell attachment and growth. Biodegradable materials used in orthopedic applications include degradable synthetic polymers and degradable metals and alloys [2]. According to Hoffman [26] dozens of different polymeric BMs have been developed to substitute metal implants for internal fracture fixation, such as bone plates, screws, and intramedullary pins. He adds that their main limitation is the loss of mechanical strength within a short time interval. On the other hand, polymeric BMs have overcome metals in some important quality attributes, such as elasticity, flexibility, longevity, and bio-inertness [27]. Among them, polyglycolide (PGA), polylactide (PLA), and polycaprolactone (PCL) have been the most widely used for this purpose due to their good biocompatibility [28]. Polymers PLA, PGA and their co-polymer compositions are most often used in applications that include fracture-fixation pins and plates, interference screws, suture anchors and other fixation implants as they are highly resorbable [29]. Implants made from PLA are used, e.g., for the surgery and/or treatment of maxillofacial fractures, ankle fractures and syndesmosis injury [30,31]. The degradable polymer poly-l-d, l-lactide (PLDLA) is applicable for the treatment of mandibular fractures, since it has good mechanical properties. The screws made from this material provided the same fixation strength as titanium screws [32]. Another synthetic polymer Poly-l-lactic (PLLA) is notable for its gradual degradation and thanks to that is applicable in orthopedics for anterior cruciate ligament reconstruction, ankle fracture treatment or meniscus injury therapy [33,34]. Bio-absorbable screws made from copolymer PLLA/PGA are suitable, e.g., for fixation of osteochondritis dissecans lesions [35]. 

Biodegradable metals are seen as promising alternatives to non-biodegradable metals. Among the metals, magnesium (Mg), zinc (Zn), and iron (Fe) are considered as materials with the most biodegradable potential [36]. During the last decades, Mg-based alloys have been intensively explored by researchers in the context of orthopedic applications. The advantages of Mg-based biodegradable metals are that their bioactivity enhances osteogenesis and that their elastic modulus matches that of bone [37,38]. Magnesium’s good properties mean that it is often used to treat bone fractures, for example in the form of an implantable screw. Typically used magnesium alloys include high-purity magnesium alloy, MgCa0.8 alloy, MgYREZr alloy and Mg-Al-Zn alloys. For example, the MgYREZr screws were applied to treat hallux valgus with good therapeutic effect [39]. Currently, Zinc-based BMs are receiving considerable attention. A comprehensive review of related research progress on Zn-based BMs for orthopedic internal fixation is presented in a study by Liu et al. [40]. Its authors point out the important fact that there is a critical need for development of BMs for fixation of fractures at heavy load-bearing bone sites where fractures occur most frequently.

## 3. Classification of the Selected Biodegradable Materials for Bone Defects and Soft Tissue Treatments

As BMs are not applicable only in orthopedics but also in other medical fields, in what follows, wider medicine branches will be considered regarding their implementation. In general, biodegradable materials are those that break down in the body and are gradually absorbed [39] and have suitable biocompatibility, including body compatibility and interface compatibility [41]. Thanks to the intensive research and development during the last decades, a wide plethora of biodegradable materials is usable for TE scaffolds and other applications. Therefore, in the first place, it is useful to distinguish their different nature and to categorize them into basic groups and subgroups. They are most widely classified as natural and synthetic materials. According to Sheikh et al. [42], the main three kinds of widely studied and clinically applied BMs are polymers, ceramics, and metals. We suggest categorizing them into six main groups and the related subgroups as shown in Figure 2.

The categorization is further used to systemize analyzed information regarding BMs into easily perceptible units—subsections of this section. To outline developmental tendencies of the defined groups of BMs, bibliometric methods will be employed. This method is frequently used for quantitative monitoring of published research. Moreover, it helps to provide a complementary classification of compared research activities and to visualize the trends in the existing literature. For this purpose, the search terms were defined by combining the following keywords: Biodegradable, material group name and “Tissue Engineering”. Data were collated by searching: (i) Web of Science, namely its sub database Web of Science Core Collection—all fields and all years; (ii) database Science Direct—all years, using the filter ’Research articles’. The two different databases were used to obtain more reliable results. The obtained data in this way are provided in descending order in Table 1. 

Subsequently, additional data were collated by searching the same terms only in Web of Science Core Collection database using the filter ’Year of publication’: 2013, 2014, …, 2022. The data obtained in this way are graphically displayed in Figure 3. 

From this graph it is clearly visible that the number of research publications during the recent decennium significantly differs among the analyzed groups of the BMs. Moreover, such visualization helps us to intuitively anticipate the trends of the future. Further subsections are devoted to briefly reviewing the individual groups and related sub-groups of BMs used for tissue engineering and regenerative medicine.

### 3.1. Biodegradable Polymers

Polymeric BMs can be either natural or synthetic in origin. Natural polymers have several advantages, for example better interaction between the implant and the cells. However, their several disadvantages—such as more difficult availability in larger quantities and more demanding processing [43], including cleaning, drying, softening, pulping—caused an increased interest in the research of synthetic polymers. Compared to natural polymers, synthetic polymers offer the possibility of adjusting their parameters [44]. Depending on the use, it is possible to adjust the mechanical properties, such as porosity or degradation rate. Synthetic polymers are also readily available and can be produced in large quantities. Polymers produced in this way have good mechanical and physical properties, such as tensile strength, modulus of elasticity and rate of degradation. 

Both subgroups of polymeric BMs, natural and synthetic, are useful to classify into classes and subclasses as it brings several benefits, e.g., it simplifies the procedure of their selection for specific applications. The following classification can be derived from the relevant literature according to their origin and method of production as shown in Figure 4.

The following two paragraphs aim to provide a brief review of both subgroups of polymeric BMs in the context of their applications in tissue engineering.

#### 3.1.1. Synthetic Polymers

The most common synthetic polymers for TE and drug delivery are aliphatic polymers that are frequently used as matrices for bioresorbable porous scaffolds for TE applications. They include: (PLA), (PGA), poly lactic-co-glycolic acid (PLGA), PCL, poly-p-dioxanone (PDS), and a copolymer of soft trimethylene carbonate and glycolide [45]. PLA exists in three forms: PLLA, poly-d-lactic acid (PDLA), and poly-d,l-lactic acid (PDLLA). Several aliphatic polymers, such as PDLLA, PLA, PGA, and PLGA have been extensively studied to assess their suitability for the treatment of patients with damaged organs or tissues and for drug delivery systems [46,47]. These polymers have been shown to be biocompatible and degrade to non-toxic products with a controllable degradation rate when implanted in vivo. Other biodegradable synthetic polymers include polyanhydrides, polyphosphazenes, polyurethanes, and synthetic hydrogels. These are applicable as sutures [48], drug delivery systems [49], artificial skin [50], wound healing [51] and orthopedic implants [52]. 

Although poly (α-hydroxyesters) such as PGA, PLA, PCL, and their copolymers degrade by hydrolysis and can be metabolized and excreted, acidic degradation products can be a potential concern for the biocompatibility of some polymeric materials, including some poly-α-hydroxyesters [53]. Therefore, this issue should be thoroughly addressed in the context of specific applications. Other polymers such as poly(ethylene succinate) (PES), and poly(butylene succinate) (PBS) are frequently applied in TE in form of the polymer itself as well as in form of composite materials [54]. In general, polymeric materials have limited strength and mechanical stability when made from large volume particles that have a macroporous structure, which is a desirable attribute for regenerative materials. These polymers undergo a process of total erosion, which can result in premature failure of the scaffolds. In addition, they are not osteoconductive and do not adequately support the adhesion, growth, and proliferation of bone cells [55]. 

Similarly, as the main groups of BMs were categorized in Table 1, the bibliometric method has been employed to evaluate the selected synthetic polymers presented in Figure 4. The search terms were defined as follows: material name and “Tissue Engineering”. Data were collated by using two independent databases Web of Science and Science Direct. The data obtained in this way are presented in descending order in Table 2.

When comparing the numbers of publications from Web of Science and Science Direct, the data demonstrates almost the same tendency, except for synthetic polymer PDS that is categorized slightly differently by both databases. 

To analyze development of research publications related to studies on the above selected synthetic polymers in TE, additional data were acquired by searching the same terms only in Web of Science Core Collection database using the filter ’Year of publication’: 2013, 2014, …, 2022. The data that was generated in this way are graphically presented in Figure 5.

From Figure 5, it can be observed that PCL is the most preferred polyester for TE application. The graph in Figure 5 also confirms the fact that PLGA and PLA are very suitable and promising for TE. Moreover, the trend in numbers of publications, shown in Figure 5, clearly corresponds with the bibliometric analysis presented in work [56] where PCL, PLGA, PLLA, PDLLA and PGA were compared during the 2005, 2006, …, 2015 using Web of Science database.

#### 3.1.2. Natural Polymers

Natural polymers include polysaccharides (e.g., starch, alginate, chitin, chitosan, cellulose) or proteins as gluten, collagen, fibrin gels, silk, and others. Moreover, biodegradable polymers derived directly from microorganisms, such as polyhydroxybutyrates (PHBs), polyhydroxyalkanoates (PHAs), and poly(3-hydroxybutyrate-co-3-hydroxyvalerate) PHBV belong to this category. Especially, PHBV material has excellent biocompatible and biodegradable properties for TE [57]. Natural polymers are widely employed in many biomedical applications, where they are used alone, in composites or blends [58]. Especially, biopolymer-based composites containing chitin, chitosan, or collagen have a good biocompatibility and biodegradability which are of utmost importance for bone TE [59]. Moreover, they also can stimulate the immune response. The molecular structure of natural polymers is highly organized and contains extracellular ligands that can bind to cellular receptors. Although naturally derived polymers are frequently used in TE, it deserves to mention their disadvantages that they are expensive and difficult to process into the desired shape when used as scaffolds in tissue engineering. The rate of degradation of both natural and synthetic polymers may vary from patient to patient, as the degradation of natural polymeric materials depends on enzyme activity, which is variable among patients [43]. A very common natural polymer is collagen. At least 28 different types of collagens are currently recognized [60]. Collagen is found in the connective tissues of mammals, which gives the tissues strength and elasticity. The most abundant is collagen type I, which is abundant in tissues, with higher levels found in tendons, skin, bones, and fascia. This material is also usable as a possible membrane barrier in guided tissue regeneration surgery [61]. Moreover, it can be applied in surgery as a suture material in the form of tendons [62] and for the purpose of drug delivery [63]. Chitin and chitosan are widely used in biomedical applications such as TE, drug delivery, and wound healing. Chitosan, which is derived from chitin by deacetylation, has the potential of forming gels [64], is a very good viscosity-enhancing agent in an acidic environment [65], is complete biodegradable, and has antibacterial, anti-tumor, and antioxidant properties [66]. As known, scaffold for tissue regeneration is made from a wide range of potential sources, from plastics to proteins. Gelatin, which is denatured and hydrolyzed form of collagen represents a promising material for scaffold engineering especially for 3D cell culture with therapeutic and regenerative properties [67]. The other important natural polymers were recently comprehensively reported in context of bone tissue engineering in work of Guo et al. [68]. Analogically, as the selected synthetic polymers were categorized in Table 2, the bibliometric method was used to evaluate the selected natural polymers presented in Figure 4. The search terms were determined in the following manner: ’material name’ and “Tissue Engineering” by using databases Web of Science, and Science Direct. The retrieved data are presented in descending order in Table 3.

The data in Table 3 show that the numbers of publications from Web of Science and Science Direct have almost the same tendency, except for polymer PHA that is categorized non-uniformly by both databases. 

Subsequently, an additional data was retrieved from Web of Science Core Collection database by searching the same terms only by using the filter ’Year of publication’: 2013, 2014, …, 2022. The data obtained in this way are graphically presented in Figure 6.

The graph in Figure 6 indicates that natural polymers extracted from biomass attract more attention from researchers than natural polymers produced by microorganisms. Moreover, the subgroup of natural polymers is a more frequent subject of research studies than the subgroup of synthetic polymers.

### 3.2. Biodegradable Composites

The strength and workability of polymers can be supplemented by the excellent bioactivity of bioglasses by making composite materials [69]. Organic-inorganic (O-I) composites made of bioglasses, and biodegradable polymers are an advantageous material due to the possibility of combining their properties and the possibility of obtaining the desired mechanical properties, biodegradability and bioactivity. Typically, these composites are prepared by using polymers as matrices and bioglass powders as fillers. Composite scaffolds made of bioglasses and polymers such as PCL, PLA, PGA, PLGA, etc. have shown better mechanical properties compared to pure bioglasses or pure polymers [70].

The preparation of composites from bioglasses and bioresorbable polymers also modifies the degradation behavior of the polymers. Acidic degradation by-products of polymers can be toxic to cells, while bioglass degrades by releasing cations that can buffer acidic by-products and maintain a neutral pH at O-I interfaces [71]. Bioglasses are hydrophilic and the inclusion of bioglasses in hydrophobic polymer matrices also changes the surface and overall properties of organic-inorganic composite scaffolds by increasing hydrophilicity and water absorption and thus modifying the degradation kinetics [72]. However, it is difficult to match the degradation rates of these two components in organic-inorganic composites [73]. Ideally, both the polymer phase and the bioglass phase should be degraded synchronously and at an adequate rate so that the scaffolds can be gradually replaced by newly formed tissue and at the same time maintain their mechanical integrity to support and control bone regeneration. In conventional O-I composites, different phases degrade at different rates, which causes uneven dissolution and mechanical instability of the scaffolds. An alternative way to overcome these non-uniform properties is the production of O-I nanocomposites in which inorganic nanoparticles or nanofibers are mixed with a polymer matrix [74].

### 3.3. Materials Based on Extracellular Matrix

Another biodegradable material with potential in the field of regenerative medicine is the native extracellular matrix (ECM). Each tissue type has a specialized structure and composition of the extracellular mass that modulates cellular responses and favors cell survival in that tissue [75]. The extracellular mass consists of two main components—collagen and preteoglycans, which are secreted by cells and are arranged specifically according to the type of tissue [76]. It also contains growth factors and cytokines that send signals regulating cell proliferation and migration, while also modulating cell differentiation and phenotypic expression. Thanks to its properties, the use of tissue-specific cell mass in the field of tissue regeneration is on the rise [76]. 

Bone extracellular mass consists of organic and inorganic parts. The organic part, mostly composed of type I collagen, provides elasticity to the tissue. The inorganic part consisting mainly of calcium phosphate is the source of bone strength [77]. In addition, there are four types of cells in bone tissue that contribute to osteogenesis: (a) undifferentiated osteoprogenitor cells, (b) mass-depositing osteoblasts, (c) mature osteocytes that no longer deposit mass, and (d) bone-resorbing osteoclasts. In natural tissue maintenance as well as in response to injury, these cell types work together to homeostatically build and break down mass [78]. 

Before using the extracellular mass in regenerative therapy, the harvested tissue must undergo a decellularization process. Decellularization is a tissue modification process in which cellular parts are removed, resulting in a non-cellular extracellular mass that can be used for therapeutic applications. The choice of a given decellularization method depends on the type of tissue. The main advantage of decellularized extracellular mass (dECM) is that it preserves the components of the natural cellular environment [76]. With proper decellularization, the complex biomolecular and physical cues in the extracellular mass are preserved and can promote cell growth. Nanofiber scaffolds based on ECM for articular cartilage regeneration is one of the promising applications of this material [79]. 

### 3.4. Biodegradable Metals

Research in the field of biodegradable metal materials is currently on the rise. Three main groups of biodegradable metals are investigated: magnesium-based, zinc-based, and iron-based (Fe-based). Biodegradable magnesium-based metals are at the forefront of research into biodegradable medical implants [2].

#### 3.4.1. Magnesium-Based Alloys

Magnesium alloys have good mechanical properties and biocompatibility. Several Mg alloying systems have been developed (for example: Mg-Ca, Mg-Sr, Mg-Zn, Mg-Si). A lot of research has been done on microstructures, mechanical properties, degradation behavior, in vitro and in vivo animal biocompatibility studies, and clinical trials to see if they can be used for biomedical purposes. Pure magnesium is known to corrode quickly [80], but the rate of corrosion is greatly reduced when purity is increased through purification. The proportion of impurities in magnesium has a significant impact on its corrosion rate and therefore grain coarsening in Mg alloys is important. Magnesium grains are coarsened by heat treatments like forging or rolling [81], while calcium is used for grain refining in magnesium alloys [82]. 

The biodegradation rate of magnesium-based alloys is controlled in several ways, for example by the choice of alloying elements, by microstructural adjustment or by surface modification. Ultrafine-grained structure of Mg-based biodegradable metals obtained by rapid solidification (RS) [83] or severe plastic deformation techniques (SPD), such as the equal channel angular pressing (ECAP) [84], or cyclic extrusion and compression (CEC) [85] have positive effect on mechanical properties and corrosion resistance. Another strategy employed for controlling the degradation rate of magnesium alloys is through coating their surfaces with calcium phosphate [86], polymer coatings, fluorinated coatings, etc. A comprehensive review focused on surface treatment techniques to control the corrosion rate and surface integrity of Mg-based alloys is presented in study [87]. Magnesium-based implants are presently used as micro clips for laryngeal microsurgery, orthopedic and cardiovascular systems.

#### 3.4.2. Iron-Based Alloys

Research in this area is focused on the parameters of the degradation of Fe and its alloys in the human body [88]. Due to the slow degradation of iron and its alloys (the degradation rate of pure Fe in an osteogenic environment is 0.16 mm/year [89]) and their ferromagnetic nature, implants based on this are considered problematic for permanent applications [90]. For this reason, alloying elements such as Mn, C, Si and Pd are usually added to these alloys to increase the rate of their degradation and reduce their magnetic ability. In addition, attention is also paid to increasing the surface bioactivity of these materials, which have been found unable to stimulate bone formation, similarly to bioinert anti-corrosion steels [91].

#### 3.4.3. Zinc-Based Alloys

Zinc, which is classified as a post-transition metal, represents the human body’s second most abundant trace element [92]. Although it is not present in large quantities but still plays critical biological roles. The degradation rate of Zn-based alloys is moderate and falls between the rates of Mg and Fe biodegradable metals [93]. One of the disadvantages of pure Zn as potential biodegradable metal lies in that pure Zn has quite low strength and plasticity. Appropriate ways to achieve a modification of mechanical properties of pure Zn would be adding alloying elements and performing microstructural adjustment [94]. Zn-Mg binary alloys were found to have enhanced tensile strength and micro hardness when compared against pure Zn [95]. Regarding the possible uses of Zn based alloys recent research shows that cardiovascular applications prevail.

### 3.5. Bioceramics

There has been important progress in the development of bioactive ceramics, glasses and glass ceramics from the second half of the 20th century. In the 1960s was initiated research on the carbonate-substituted calcium phosphates, and since then the research on these materials is mainly focused for their use in bone and dental tissue engineering [96]. Especially bioceramics such as calcium phosphates, and bioactive glasses have been widely used for bone regeneration and replacement and TE applications.

#### 3.5.1. Calcium Phosphates

Synthetic calcium phosphates (CaP) are osteoconductive and bioabsorbable. Moreover, they are similar to the inorganic component of bone. Calcium phosphates used for bone repair are classified according to their composition into the following groups [97]: (i) Calcium deficient apatite (CDA), (ii) Hydroxyapatite [HA, Ca_10_(PO_4_)_6_(OH)_2_], (iii) Beta-tricalcium phosphate [β-TCP, Ca_3_(PO_4_)_2_], (iv) Biphasic calcium phosphate (BCP)—A mixture of HA and β-TCP with different weight ratios of HA/β-TCP. According to Xiao et al. [98], HA, β-TCP, a BCP are the most explored natural ceramics for bone regeneration.

The production of dense CaP scaffold for bone regeneration requires sintering at temperatures of 1000 to 1200 °C. Degradation of CaP in vitro or in vivo depends on their composition, physical shape, crystallinity, porosity, and preparation conditions [99]. The bioactivity of CaP bioceramics was observed by direct attachment to native bone on a biomaterial surface coated with hydroxyapatite [100]. The formation of biomimetic carbonate apatite on CaP surfaces in simulated body fluid (SBF) was also demonstrated as an in vitro bioactivity by uptake of calcium and phosphate ions from the solution [101,102]. CaPs allow osteoblastic cells to attach, proliferate and differentiate [103]. Differentiating osteoblast cells seeded on BCP produce collagen type I, alkaline phosphatase, proteoglycans (decorin, lumican, biglycan) and bone proteins (osteocalcin, osteopontin and bone sialoprotein) known to be expressed in bone formation [104]. CaP coatings on bioinert materials used for total joint arthroplasty have demonstrated improved osseointegration at the bone/implant interface leading to better implant stability [105]. Ectopic bone formation in vivo has also been demonstrated when CaP-coated implants were placed in non-bony sites [106].

#### 3.5.2. Bioactive Glasses

Bioactive glasses belong to the class of non-crystalline silicate glasses that can stimulate the formation of bone-like minerals (hydroxide carbonate apatite, HCA) in the presence of physiological fluids. HCA is like the inorganic component of natural bone, and the HCA layer is thought to interact with the bone extracellular matrix (ECM) to fuse with natural bone [107]. Bioactive glass (BG) consists of 46.1 mol% SiO_2_, 24.4 mol% Na_2_O, 26.9 mol% CaO, and 2.6 mol% P_2_O_5_, later it was named as Bioglass^®^, which during in vivo studies formed a strong bond with native bone due to the formation of an HCA layer at the bone/implant interfaces, followed by dissolution of glass materials [108]. Several types of bioactive glasses were gradually developed–silicate-based, phosphate-based, and borate-based [109]. 

Phosphate-based bioglasses have a chemical affinity for bone due to their similarity to the inorganic phases of bone. This group of bioglasses have a high dissolution rate in aqueous media due to the easy hydration of the P–O–P bond [110,111]. The rate of dissolution can be influenced by adding suitable metal oxides to the composition of the glass, e.g., TiO_2_, CuO, NiO, MnO, Fe_2_O_3_. For this reason, phosphate-based bioglasses have been investigated in tissue engineering as carriers of antibacterial ions with their controlled release [112]. 

In the development of artificial bone tissue, it is necessary to adapt the degradation rate of the biomaterial scaffold. Modifying the composition of bioglass makes it possible to control its degradation rate in vitro and to increase bone regeneration. For example, in silicate-based bioglasses, by partially replacing SiO_2_ with B_2_O_3_, the degradation rate can vary over a wide range [113]. In this way, it is possible to match the rate of degradation of borate-based bioglasses with the rate of formation of new bone extracellular mass. Borate-based bioglasses promote cell proliferation along with differentiation in vitro, while in vivo studies reported that boron increases tissue infiltration [114,115]. Borate-based bioglasses are promising candidates for tissue engineering applications, also for their properties such as bioactivity and osteoconductivity.

### 3.6. Nanocomposites

O-I nanocomposites prepared with nanoparticle bioglass filler provide a larger surface area compared to conventional composites prepared with microscopic bioglass particles. The increased surface size of bioglass positively affects the interactions between cells and the material. Bioceramic nanoparticles improved protein adsorption and osteoblast adhesion compared to their micro-particulate counterparts [116]. To obtain organic-inorganic nanocomposites with improved bioactivity, cell-material interactions and mechanical properties, the size of nanoparticles is an important parameter. In a detailed study on porous 3D PLLA/bioglass nanocomposite scaffolds, it was observed that the addition of bioglass nanoparticles up to 20% of weight did not change their morphology but increased their bioactivity [117,118]. As the amount of bioglass increased from 0 to 30% of weight, the pressure modulus in the nanocomposite scaffolds increased from 5.5 MPa to 8.0 MPa. The incorporation of bioglass nanoparticles into the PLLA matrix also helped to balance water uptake by the nanocomposite scaffolds and affected the rate of degradation. Bioglass nanofibers have also been used to fabricate nanocomposite scaffolds. These nanocomposites induced osteoblast-like cell attachment, spreading and proliferation in vitro. In general, O-I nanocomposites prepared with bioactive glass nanoparticles or nanofibers showed better mechanical properties and cell-material interactions compared to conventional micro-composites due to their higher surface area to volume ratio [97].

## 4. Smart Biodegradable Materials for Tissue Engineering

Traditionally, biodegradable materials were designed to interact with living tissue temporarily or permanently to provide functions, such as mechanical support. Smart BMs are defined as those that respond to external stimuli, such as light, magnetic fields, ultrasound, etc. Typical smart BMs include, e.g., photoresponsive and chemoresponsive polymers that combine sensing and actuation within the same material, without need for external devices [119,120,121]. Moreover, development of smart bioactive glasses for bone contact applications is becoming a hot research area in TE [122]. Recent research in this domain focuses on the molecular interaction of bioactive glass-based ionic dissolution products with their physiological surrounding environment [123]. Another example of smart biomaterial is decellularized extracellular matrix, which is the noncellular component of tissue that retains relevant biological cues for cells [124]. The related research is oriented towards directly using the component of the dECM to obtain scaffolds simulating native ECM [125]. 

Montoya et al. [126] suggested classifying smart biomaterials according to their degree of interaction with their external environment and the subsequent biological responses to clarify the concept of smartness in this context. The authors categorized smart materials into three kinds, namely, active, responsive, and autonomous. The inert biomaterial is just biocompatible or bioinert, while the active one can provide planned one-way interaction, e.g., bioactive therapy, with biological tissue. One of the first materials of this category was bioactive glass composed of four oxides, namely SiO_2_-CaO-Na_2_O-P_2_O_5_, introduced by Hench [127]. The main limitation of active biomaterials lies in the limited duration and efficacy of the therapy due to their degradation in a biological environment. Active biomaterials, namely polymer and lipid-based ones are often used for the controlled release of drugs like antibiotics, antiseptics, vitamins, and statins [128,129]. Responsive biomaterials can receive a stimulus and provide feedback to it through triggered reactions. Examples of such materials are artificial cells and hydrogels [130]. Especially, the need for biodegradable hydrogels in biomedical applications is significant since their physical properties can be designed to follow those of articular cartilage [131]. Recent developments in the design of responsive nanocomposite hydrogels increase their potential in biomedical applications including their utilization as therapeutic platforms for the delivery of precisely prescribed medications [132]. The responsive functionalities of biomaterials can be triggered from internal or external sources. The stimuli coming from inside an organism are internal, while external sources generate stimuli from outside of the body like heat, light, chemicals, or pressure. Both kinds of the signals can be categorized into three groups: biological, chemical, and physical [133]. For instance, PLLA-based biomaterials processed into piezoelectric structures can be engineered as scaffolds for promoting cellular growth during electrostimulation [134]. The low piezoelectric effect of PLLA is similar in magnitude to that of natural biomacromolecules like collagen [135] giving it the ability to interact with biological systems without being rejected [136]. The highest degree of smartness represents biomaterials capable of autonomously responding to the surrounding environment. Biomaterials with such properties can be considered as kind of dynamic biomaterials, which respond to stimuli by autonomous feedback loops [130]. The model of autonomous biomaterial is graphically illustrated in Figure 7.

Smart biomaterials can be applied, e.g., for the regulation of stem cell activity, as well as to understand complex cellular processes [137]. The control of dynamic biomaterials after implantation in the body becomes challenging research in this field. For this purpose, Badeau et al. [138] employed a logic-based peptide hydrogel as a miniature computer system taking inputs from the surrounding microenvironment to decide when to release therapeutic agents for drug delivery. Research devoted to smart biomaterials in biomedical engineering is widely published, and its development is comprehensively summarized in recent works [139,140,141].

## 5. Conclusions

This article aimed to approach the issue of biodegradable materials especially from the perspective of their development and classification. One of its intentions was also to point out the typical application areas of BMs in orthopedic practice. As a result, the clinical indications for the use of biodegradable implants in orthopedics were comprehensively identified and graphically presented. To characterize development tendencies of BMs, bibliometric analysis has been employed for estimating the research trends among the main groups of the materials and to sort them based on the frequency with which the keywords occur in publications in recent ten years. Results showed that the largest research attention is given to polymers, composites and ECM based materials. Because polymers have been identified as the dominant group among BMs, specific types of frequent polymers were selected and categorized using the proposed classification framework and arranged from the viewpoint of the number of publication outputs. One can see that among the synthetic polymers, PCL is mostly used in the context of TE research.

Among natural polymers, primarily collagen, has long been used in biomedical applications for implants and tissues injuries. Moreover, the bibliometric analysis showed a constantly increasing trend of chitosan, gelatin, cellulose and PCL among biodegradable polymers. 

When analyzing current challenges in the given research field, it is useful to emphasize the fact that successful application of the biodegradable materials for the specified purposes requires the possibility of adapting their mechanical properties and a rate of degradation that is compatible with the rate of formation of new tissue. In addition, factors that are beneficial for cell growth and proliferation, such as high porosity, are in conflict with good mechanical properties. Moreover, the individual types of described biodegradable materials have different advantages and disadvantages that have not yet been comprehensively analyzed and rigorously evaluated. Consecutive research could be focused on the outlined issues. As regards further the literature research on biodegradable materials for TE, it could be oriented towards mapping and classifying biodegradable composites from different viewpoints, for example, to classify them according to their functionality.

## Figures and Tables

**Figure 1 jfb-14-00159-f001:**
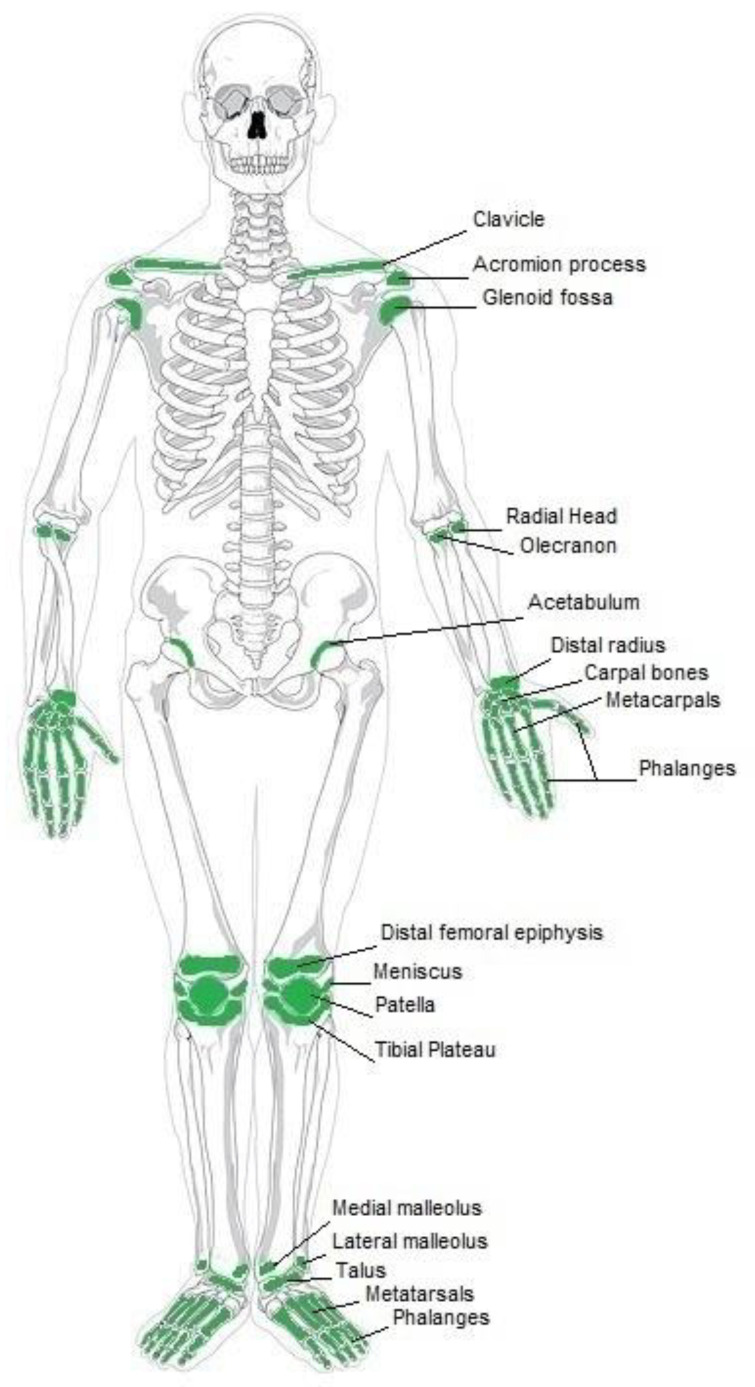
Current clinical applications of biodegradable implants.

**Figure 2 jfb-14-00159-f002:**
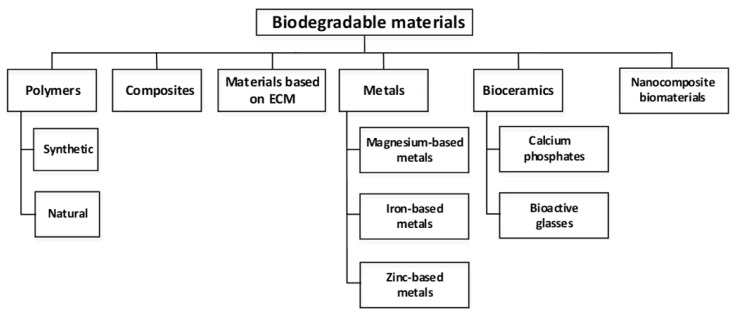
Main groups and subgroups of the BMs.

**Figure 3 jfb-14-00159-f003:**
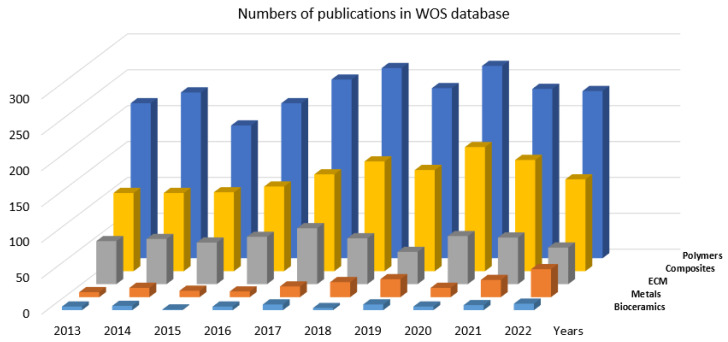
Bibliometric comparison of the main groups of BMs.

**Figure 4 jfb-14-00159-f004:**
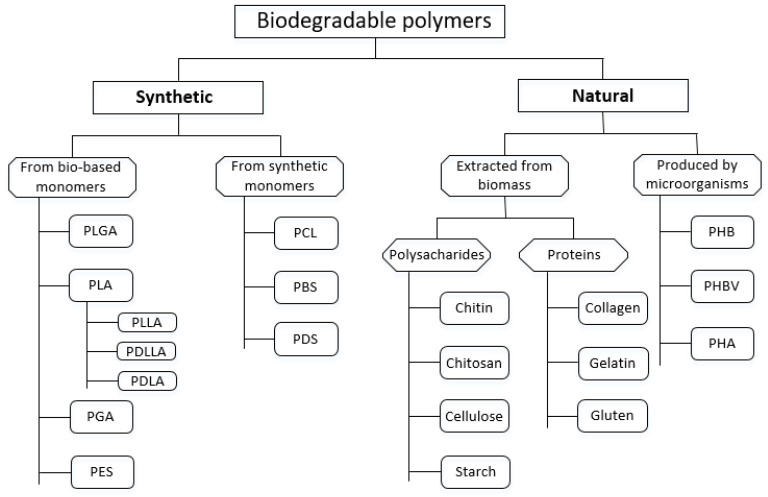
Classification of polymeric BMs according to their origin and method of processing.

**Figure 5 jfb-14-00159-f005:**
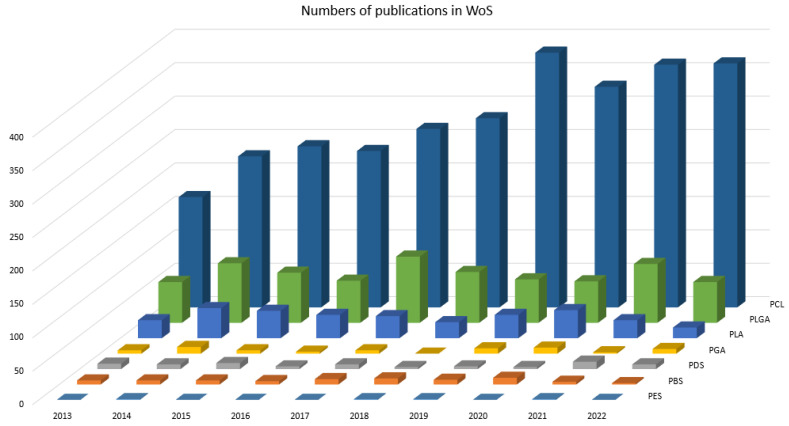
Mutual comparison of the selected synthetic polymers.

**Figure 6 jfb-14-00159-f006:**
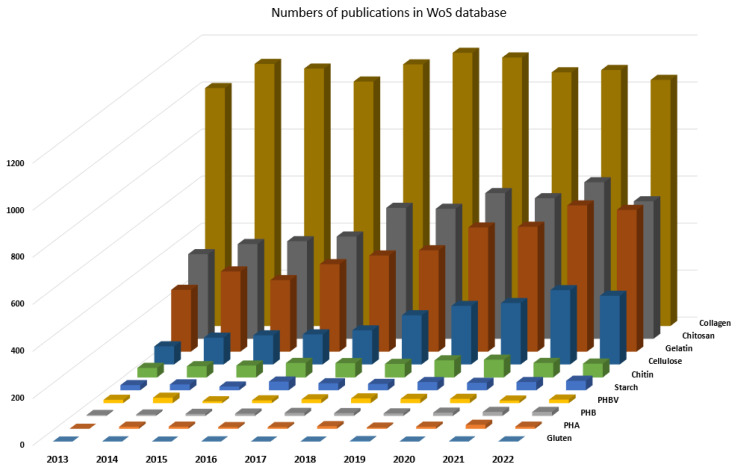
Bibliometric comparison of the selected natural polymers.

**Figure 7 jfb-14-00159-f007:**
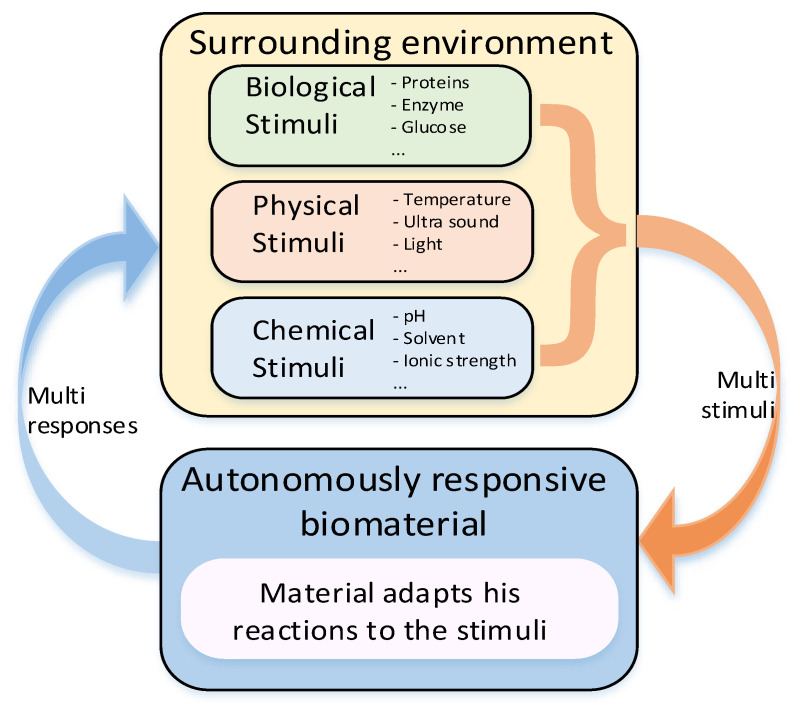
Model of stimuli-responsive biomaterial with closed loop.

**Table 1 jfb-14-00159-t001:** Numbers of research publications related to studies on BMs in tissue engineering.

Search Terms	Number of Publications (All Years)	
	Web of Science Core Collection, All Fields	Science Direct, Research Articles
Biodegradable Polymers “Tissue Engineering”	4640	17,973
Biodegradable Composites “Tissue Engineering”	2076	9655
Biodegradable extracellular matrix “Tissue Engineering”	1275	8560
Biodegradable Metals “Tissue Engineering”	208	5705
Biodegradable Bioceramics “Tissue Engineering”	105	1275

Note: All the data were retrieved on 30 January 2023.

**Table 2 jfb-14-00159-t002:** Numbers of research publications related to studies on BMs in tissue engineering.

Abbreviations	Search Terms	Number of Publications (All Years)	
		Web of Science Core Collection, All Fields	Science Direct, Research Articles
PCL	“Polycaprolactone” “Tissue Engineering”	3624	4136
PLGA	“Poly Lactic-co-Glycolic Acid” “Tissue Engineering”	1122	1917
PLA	“Poly Lactic Acid” “Tissue Engineering”	629	1451
PGA	“Poly Glycolic Acid” “Tissue Engineering”	166	898
PDS	“Polydioxanone” “Tissue Engineering”	100	196
PBS	“Poly(butylene succinate)” “Tissue Engineering”	84	229
PES	“Poly (ethylene succinate)” “Tissue Engineering”	4	225

Note: All the data were retrieved on 30 January 2023.

**Table 3 jfb-14-00159-t003:** Numbers of research publications related to studies on natural polymers in tissue engineering.

Search Terms	Number of Publications (All Years)	
	Web of Science Core Collection, All Fields	Science Direct Research Articles
Collagen “Tissue Engineering”	18,222	20,894
Chitosan “Tissue Engineering”	6950	10,494
Gelatin “Tissue Engineering”	5595	8951
Cellulose “Tissue Engineering”	2218	6984
Chitin “Tissue Engineering”	867	3127
Starch “Tissue Engineering”	478	2319
“Poly(3-hydroxybutyrate-co-3-hydroxyvalerate)” “Tissue Engineering”	234	375
Polyhydroxybutyrate “Tissue Engineering”	174	268
Polyhydroxyalkanoate “Tissue Engineering”	117	532
Gluten “Tissue Engineering”	12	122

Note: All the data were retrieved on 30 January 2023.

## Data Availability

Not applicable.

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
