# Peer review of "Biodegradable Materials for Tissue Engineering: Development, Classification and Current Applications"

_jfb, 2023, doi:10.3390/jfb14030159_

Round 1

Reviewer 1 Report

Line 55: This is not entirely correct, as biodegradable implants can also be used for the stabilization of other types of bones, such as cortical bones.

Line 60: However, this statement is not entirely correct as biodegradable implants are not typically used for the treatment of all types of fractures in the tibia and fibula. The use of biodegradable implants for fracture treatment in these bones depends on various factors, including the location and severity of the fracture.

Line 86: "causes that" should be "causing"

Line 86-87: unclear. However, their use can prevent bone ingrowth into the implant and limit its ability to adapt to changes in bone structure?

While it is true that non-degradable metals can cause a lack of bone ingrowth, this is not a result of the material itself but rather the design of the implant. Non-degradable metals can be designed to promote bone ingrowth by incorporating porous surfaces or coatings that promote bone cell attachment and growth.

Line 108: it should be better to say, "Biodegradable metals are seen as promising alternatives to non-biodegradable metals." 

Line 146: a typo in line where "by" should be "be".

Line 188: The verb "is" should be replaced with "are"

Line 207-209: While it is true that these polymers have been studied for tissue engineering applications, they have not been approved for clinical use in patients with damaged organs or tissues!

Line 215-216: While it is true that acidic degradation products can affect the biocompatibility of some polymeric materials, it is not accurate to say that this is always the case with poly (α -hydroxyesters) such as PGA, PLA, and PCL. Biocompatibility is a complex concept that depends on various factors, including the specific application, the type and amount of degradation products, and the host response. It would be more accurate to say that acidic degradation products can be a potential concern for the biocompatibility of some polymeric materials, including some poly (α -hydroxyesters), and that this issue should be carefully evaluated in the context of specific applications.

Line 234: misspelling of "data" as "date" 

Line 305: This statement implies that all scaffold materials can come from plastics or proteins, but this is not the case, as scaffolds can also be made from other materials, such as ceramics. 

Line 477: Zinc is not a transition metal but a post-transition metal. Additionally, this sentence could be more accurate by stating that zinc is the human body's second most abundant trace element. It is not present in large quantities but still plays critical biological roles.

Line 488: "initiated" should be replaced with "initiated the".

Line 523: % SiO2 repeated twice.

Line 526-527: state that "several types of bioactive glasses were gradually developed – silicate-based, phosphate-based, and borate-based." However, in the next sentence, it says that there are "phosphate-based bioglasses" and "borate-based bioglasses," which suggests that these are specific types of bioactive glasses rather than just categories.

Line 545:  "interact with living tissue temporarily" should be "interact with living tissue temporarily or permanently" to reflect the fact that some biodegradable materials are designed to provide long-term support.

Line 546: "Smart BMs are thought as" should be "Smart BMs are defined as."

Line 556: "Montoya et al suggested to classify" should be "Montoya et al suggested classifying."

Line 559: "They distinguish among three kinds of smart biomaterials" should be rephrased

Line 565: "degradability in a bioactive environment" should be "degradation in a biological environment."

Line 581: "piezoelectric effect of PLLA is comparable to the piezoelectricity of natural biomacromolecules of collagen" should be "piezoelectric effect of PLLA is similar in magnitude to that of natural biomacromolecules like collagen."

Line 616-617: "biomedical applications for biomedical applications" is repeated twice.

Reviewer 2 Report

This review paper offers a broad overview of biodegradable materials utilized in regenerative medicine, encompassing polymers and inorganic composite materials. While not a comprehensive analysis of any one topic, it can provide valuable insight into the development trends and distribution of biodegradable materials.

However, there are a few areas that could benefit from improvement. Firstly, it may be more appropriate to discuss biodegradable nanocomposites last, after outlining other inorganic materials like metals and bioactive glass. Additionally, instead of simply categorizing composites as conventional or nanocomposites, it would be more informative to classify them according to their functionality. For instance, research on additives such as growth factors or conductive materials like graphene and biodegradable metals that promote regeneration should be more widely explored.

Overall, this review paper is a valuable resource for those interested in the utilization of biodegradable materials in regenerative medicine, but further improvements could enhance its overall effectiveness.

Author Response

Response to reviewer 2

We would like to thank you for your very valuable comments and recommendations.

This review paper offers a broad overview of biodegradable materials utilized in regenerative medicine, encompassing polymers and inorganic composite materials. While not a comprehensive analysis of any one topic, it can provide valuable insight into the development trends and distribution of biodegradable materials. However, there are a few areas that could benefit from improvement.

Firstly, it may be more appropriate to discuss biodegradable nanocomposites last, after outlining other inorganic materials like metals and bioactive glass.

Answer:

Thank you for this useful suggestion to reorganize categorization of the main groups and sub-groups of the biodegradable materials. Accordingly, biodegradable nanocomposites are now (in revised version of the manuscript) discussed in separate sub-section 3.6 (instead of sub-sub-section 3.2.2), after describing inorganic materials like materials based on ECM, metals and bioceramics.

Moreover, this change is now also reflected in Figure 2, where nanocomposites are represented as a separate category of biodegradable materials.

Additionally, instead of simply categorizing composites as conventional or nanocomposites, it would be more informative to classify them according to their functionality. For instance, research on additives such as growth factors or conductive materials like graphene and biodegradable metals that promote regeneration should be more widely explored.

Answer:

It is a good idea to classify biodegradable composites according to their functionality, since it could bring new information to readers. However, biodegradable composites comprise a wide range of at least two-phase hybrid materials. In our opinion, to classify them according to their functionality would mean to conduct a new intensive literature research. Therefore, we used this idea as a recommendation for future research (please see a last sentence in Conclusions section of the revised version of the manuscript).

Overall, this review paper is a valuable resource for those interested in the utilization of biodegradable materials in regenerative medicine, but further improvements could enhance its overall effectiveness.

  1. 03. 2023

Reviewer 3 Report

I was pleased to review article ID jfb-2261178 entitled “Biodegradable Materials for Tissue Engineering: Development, Classification and Current Applications” for the Journal of Functional Biomaterials. The review addresses the current state of biodegradable materials that are used in tissue engineering in several applications. Overall, the article was really well written and brought the reader an easy understanding of the topic.

Author Response

Response to reviewer 3

I was pleased to review article ID jfb-2261178 entitled “Biodegradable Materials for Tissue Engineering: Development, Classification and Current Applications” for the Journal of Functional Biomaterials. The review addresses the current state of biodegradable materials that are used in tissue engineering in several applications. Overall, the article was really well written and brought the reader an easy understanding of the topic.

We would like to thank you for your time and the positive feedback on our paper.

  1. 03. 2023